# FAM9B serves as a novel meiosis-related protein localized in meiotic chromosome cores and is associated with human gametogenesis

Xin-jie Zhuang[1,2,3,4]*, Xue Feng[1,2,3,4], Wen-hao Tang[5], Jin-liang Zhu[1,2,3,4], Ming Li[1,2,3,4], Jun-sheng Li[1,2,3,4], Xiao-ying Zheng[1,2,3,4], Rong Li[1,2,3,4], Ping Liu[1,2,3,4]*, Jie Qiao[1,2,3,4]

1 Center for Reproductive Medicine, Department of Obstetrics and Gynecology, Peking University Third Hospital, Haidian District, Beijing, PR China, 2 Key Laboratory of Assisted Reproduction, Peking University, Ministry of Education, Haidian District, Beijing, PR China, 3 Beijing Key Laboratory of Reproductive Endocrinology and Assisted Reproductive Technology, Haidian District, Beijing, PR China, 4 National Clinical Research Center for Obstetrics and Gynecology, Peking University Third Hospital, Haidian District, Beijing, PR China, 5 Department of Urology, The Third Hospital of Peking University, Beijing, China

* Bysylp@sina.com (PL); zhuangxinjie902@163.com (XJZ)

**Data Availability Statement:** All relevant data are within the manuscript and its Supporting information files.

## Abstract

Meiosis is a complex process involving the expression and interaction of numerous genes in a series of highly orchestrated molecular events. Fam9b localized in Xp22.3 has been found to be expressed in testes. However, FAM9B expression, localization, and its role in meiosis have not been previously reported. In this study, FAM9B expression was evaluated in the human testes and ovaries by RT-PCR, qPCR, and western blotting. FAM9B was found in the nuclei of primary spermatocytes in testes and specifically localized in the synaptonemal complex (SC) region of spermatocytes. FAM9B was also evident in the follicle cell nuclei and diffusely dispersed in the granular cell cytoplasm. FAM9B was partly co-localized with SYCP3, which is essential for both formation and maintenance of lateral SC elements. In addition, FAM9B had a similar distribution pattern and co-localization as γH2AX, which is a novel biomarker for DNA double-strand breaks during meiosis. All results indicate that FAM9B is a novel meiosis-associated protein that is co-localized with SYCP3 and γH2AX and may play an important role in SC formation and DNA recombination during meiosis. These findings offer a new perspective for understanding the molecular mechanisms involved in meiosis of human gametogenesis.

## Introduction

Gametogenesis is a highly complex process in which gametes are produced from germinal stem cells by mitotic and meiotic cell division [1]. During gametogenesis, synaptonemal complex (SC) formation may play a universal role in meiosis. The SC is a complex protein structure with many proteins involved in its formation [2]. During this process, a large number of genes

**Funding:** This study was supported by the National Natural Science Foundation of China (NO. 8217060945, 81671513 and 81200466) and Beijing Natural Science Foundation (NO.7172236).

**Competing interests:** The authors have declared that no competing interests exist.

**Abbreviations:** aa, amino acid; CEs, central element; FITC, fluorescein isothiocyanate; LEs, lateral elements; MI, metaphase I; PFA, Paraformaldehyde; RT-PCR, reverse transcription polymerase chain reaction; SCs, synaptonemal complex; TFs, transverse filaments; γH2AX, phosphorylated histone H2AX.

participate in meiosis. However, human meiosis is still poorly understood on a molecular level.

Many important proteins participate in the assembly of SC, such as SYCP1 [3, 4], SYCE1 [5, 6], SYCE2 [7, 8], TEX12 [9], SYCE3 [10], SYCP2 [11, 12], SYCP3 [13, 14], and SLX2 [15, 16]. SYCP1 is an extended filamentous protein that consists of three domains. It is probably a TF constituent and a major protein component of SC [3]. SYCP1 is associated with infertility in both sexes and leads to synapsis failure in mice [17]. SYCE1 encodes an SC protein that plays an essential role during meiosis. Deleterious SYCE1 mutation is associated with male infertility [6, 18] and autosomal recessive primary ovarian insufficiency [19]. SYCE2 is required for SC assembly [8] and forms a highly stable, constitutive complex within the central element (CE) of the SC [7]. SYCE3 is localized in the CEs and *Syce3*$^{-/-}$ in mice and is required for male and female fertility [10]. SYCP2 is a component of metazoan SC [20], which is essential for proper chromosome synapsis [11, 21]. SYCP3 is important for SC formation, while SYCP3 mutation plays a role in male fertility [22] and recurrent pregnancy loss [23]. Female germ cell aneuploidy [24] and embryo death have been observed in mice lacking the meiosis-specific protein SYCP3 [25]. SLX2 is also localized in SC and may be involved in SC formation during spermatogenesis [16] and meiotic oocyte maturation [15].

Chromatin reorganization requires formation and repair of DNA double-strand breaks (DSBs). DSB generation rapidly results in the phosphorylation of histone H2A variant H2AX [26]. Because H2AX phosphorylation (γ-H2AX) is abundant, fast, and correlates well with each DSB, it is the most sensitive marker that can be used to examine the DNA damage produced and subsequent repair of the DNA lesion [27]. γ-H2AX appeared to be located in unsynapsed chromosomal segments during male meiotic prophase I. Several waves of H2AX phosphorylation/dephosphorylation coupled to various developmental phases of spermatogonia and spermatocytes, as well as to spermatid differentiation [28]. Previously, we found that SLX2-conserved Cor1 domain was highly homologous with SYCP3. Its co-localization with γH2AX and interaction with TIP60 suggested that SLX2 might be involved in DNA recombination and DSBs during meiosis [2]. Fam9b conserved Cor1 domain is localized in Xp22.3 and expressed exclusively in testes [29]. However, FAM9B expression, localization, and its role in meiosis remain unclear.

The present study demonstrated that FAM9B is abundantly transcribed in human testes and ovaries and is differentially expressed during human gametogenesis. Moreover, FAM9B was found in the nuclei of primary spermatocytes in testes and specifically localized in the synaptonemal complex (SC) region in spermatocytes. FAM9B was also evident in follicle cell nuclei and diffusely dispersed in the granular cell cytoplasm. FAM9B has an expression pattern similar to that of γH2AX and is specifically localized in human SC region as SYCP3 during meiosis. These results strongly indicate that FAM9B plays important roles in the regulation of SC formation during meiosis. Further clinical data analysis will provide important and novel information allowing for a further understanding of the molecular mechanism of human SC formation and meiosis.

## Materials and methods

### Sample collection and ethics statement

Twenty testicular biopsy tissue samples were obtained from patients (age: 25–36 years) with obstructive azoospermia, and 50 patients undergoing percutaneous testicular and ovarian biopsy (age: 25–36 years) for IVF treatment agreed to further tests at the Reproductive Medicine Center, Peking University Third Hospital between January 2014 and November 2017. All samples were obtained from donors (age: 25–36 years) who signed the informed consent for

excision and scientific use of testis tissue voluntarily. The present study was reviewed and approved by the Ethics Commission of the Reproductive Medical Faculty of the Peking University Third Hospital (No. 2013SZ021). All studies and protocols were approved and conducted in accordance with institutional guidelines and the Declaration of Helsinki for research involving human tissue.

## Reagents and antibodies

All chemicals and electrophoresis and transfer equipment were purchased from Sigma Chemical Co. (St. Louis, MO, USA) and Bio-Rad Laboratories (Carlsbad, CA) unless specified otherwise. Revert Aid First Strand cDNA Synthesis Kit and PCR Cloning Kits were purchased from Clontech (San Jose, CA) and Thermo Fisher Scientific Inc. (San Jose, CA, USA). Anti-SYCP3 antibody and anti-FAM9B antibody were purchased from Abcam (Cambridge, UK), while anti-$\gamma$-H2AX antibody was purchased from Upstate Biotechnology (Charlottesville, VA). Primers were synthesized by Invitrogen (Carlsbad, California, USA).

## RNA-seq and real-time PCR validation

Total RNA from samples was isolated in accordance with the manufacturer's instructions [30]. For RT-PCR, total RNA was reverse transcription using the Thermo Fisher Scientific Inc. (San Jose, CA, USA) and Superscript III reverse transcriptase as described previously [31]. Three primer pairs were used to detect Fam9b mRNA expression (Table 1). Amplification was carried out, with an annealing temperature of 58.5˚C for Fam9b primers. Reaction was finished at 72˚C for 5 min. The PCR products were analyzed using agarose gel electrophoresis. Relative amounts of cDNA were normalized against GAPDH.

## Protein extraction and western blot analysis

Western blotting and protein extraction were performed using standard protocols [15]. Human testes and ovarian biopsies were homogenized in RIPA lysis buffer. Protein concentration was determined using the Bradford Reagent (BioRad). Protein lysates were separated by SDS-PAGE. Briefly, 50 μg of total protein were loaded, separated by SDS–PAGE, and immunoblotted with anti-FAM9B antibody. Membranes were blocked in TBST incubated overnight with a 1:200 dilution of anti-FAM9B antibody (Abcam). In addition, membranes were incubated with HRP-conjugated secondary antibody and protein expression levels were measured using Image-Pro plus 5.1.

## Immunohistochemistry and immunocytochemistry

Immunohistochemistry and immunocytochemistry were performed via standard manipulations as described previously [32]. The sections were incubated with anti-FAM9B antibody

**Table 1. Primer pairs.**

| Name | Usage | Sequence (5'-3') |
|---|---|---|
| Fam9b-1 | RT-PCR | Sense: CCGCTGCCTGCAGGTTCTTTGAGG |
| | | Antisense: GACGAGTGACAGAGGA GCTATAC |
| Fam9b-2 | Q-PCR | Sense: TACATCACAGACGAGCAGAAAG |
| | | Antisense: TTCAGCCTCTCTCTCCTAACA |
| Fam9b-3 | Q-PCR | Sense: TAGGAGAGAGAGGCTGAAAGAG |
| | | Antisense: AGTTCACTGTCTTCATCGGAAA |

(diluted 1:200) or pre-immune rabbit serum as a negative control. FITC-conjugated anti-rabbit secondary antibody was used at a dilution of 1:600. The sections were then incubated with DAPI (Sigma-Aldrich) and in PBS instead of primary antibody for negative controls. Sample slides were photographed using a Nikon ECLIPSE 80i microscope with a Nikon DS-Ri1 camera (Nikon Corporation, Japan).

## Confocal fluorescence and chromosome spreading

Chromosome spreading of primary human spermatocytes was performed using the drying-down technique [15]. Briefly, human seminiferous tubules were isolated. Subsequently, the seminiferous tubules were torn into small pieces. The spermatocyte cell suspension was mixed with 3.7% PFA solution. The spermatocyte cells were subsequently spread on a clean glass slide. Then, the cell slides were washed, dried, and stained using anti-SYCP3 (1:200, Abcam), γ-H2AX (1:200, Sigma), and/or anti-Fem9b (1:200, Abcam). TRITC- or FITC-conjugated anti-rabbit secondary antibody (Jackson Laboratories, Bar Harbor, ME)) was applied at a dilution of 1:600. Protein subcellular localizations were examined using a laser confocal microscope (Zeiss).

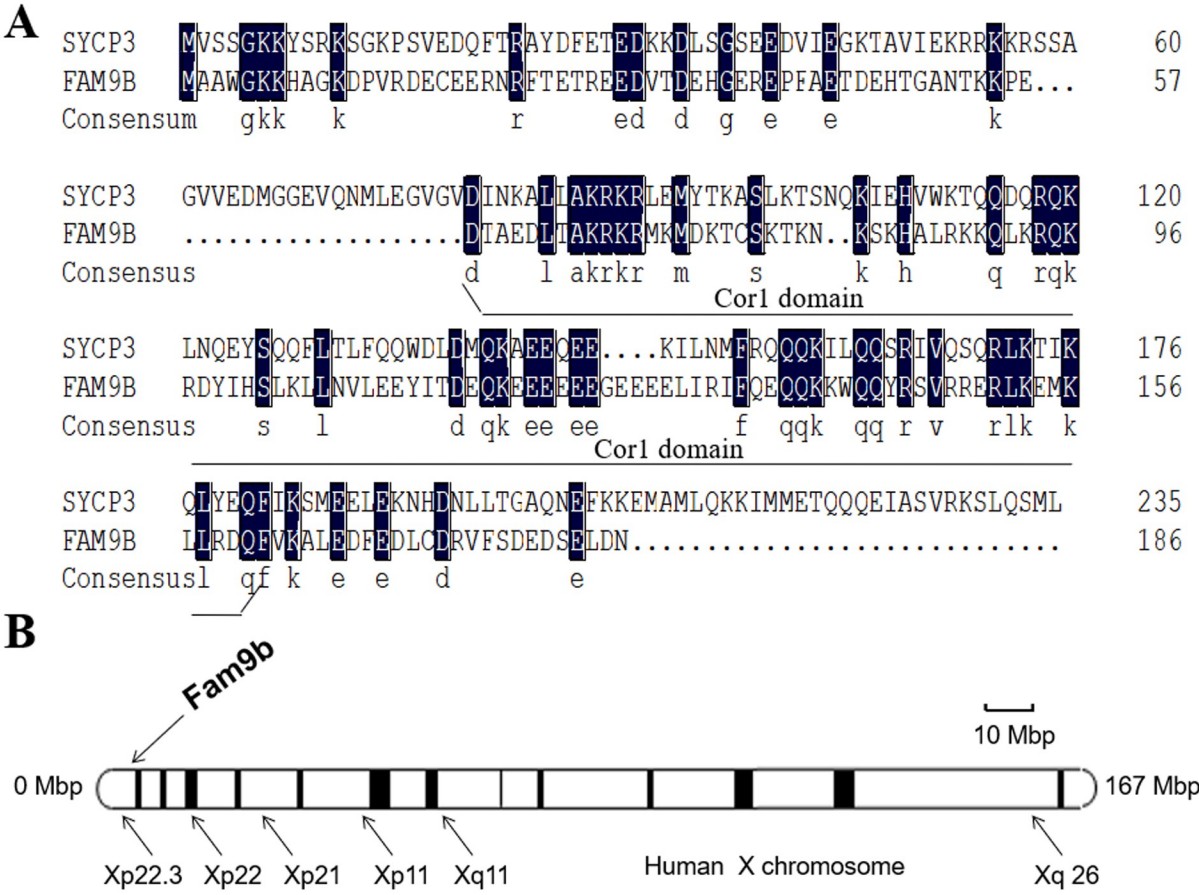

**Fig 1. Fam9 mapped on human X chromosomes shares homology with SYCP3.** (A) Fam9b -conserved Cor1 domain shares homology with SYCP3. (B) Fam9 mapped on Xp22.3 of human X chromosomes.

## Statistical analysis

The results were analyzed using SPSS 17.0 software (Chicago, IL, USA). All experiments were repeated and analyzed at least three times. An independent-sample t-test was performed on the qPCR and immunofluorescence intensity data. P-values <0.05 were considered to be significantly different.

## Results

### Fam9b is a member of Fam9 family located on chromosome Xp22.3

A FASTA comparison showed that Fam9b DNA sequences demonstrate significant homology with SYCP3, which contains a Cor1 domain (Fig 1A). Fam9b encodes 186AA and also contains a Cor1 domain. FAM9B may have functions that involve spermatogenesis as SYCP3. In addition, Fam9b is located on chromosome Xp22.3 (Fig 1B) and microdeletion/ microduplications at Xp22.3 have been frequently detected with the application of BACs-on-Beads™ and array-comparative genome hybridization technologies in prenatal diagnosis. Xp22.3 deletions

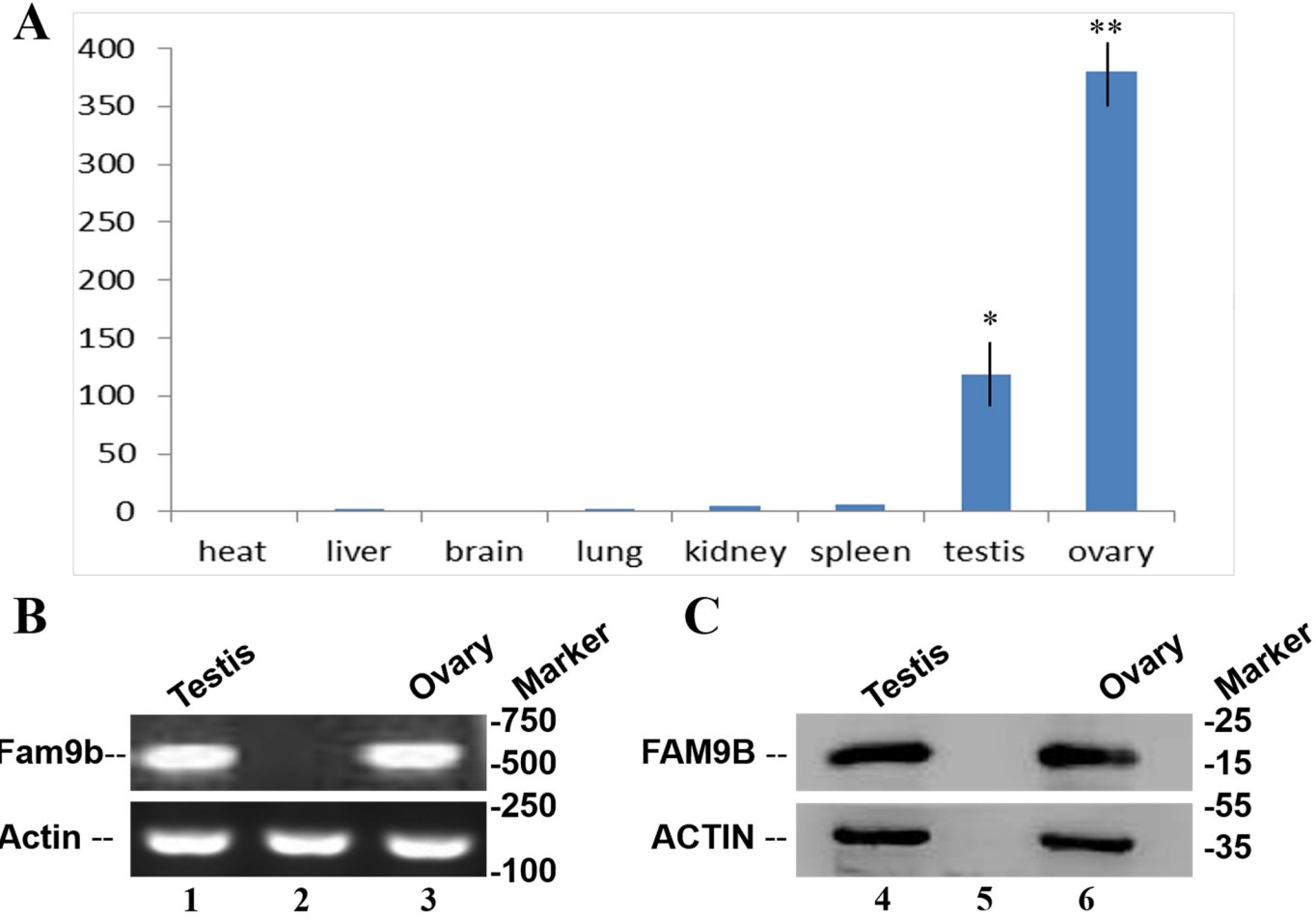

**Fig 2. FAM9B expression was determined by qPCR, RT-PCR, and western blotting.** (A) qPCR results show that FAM9B mRNAs are highly expressed in testes and ovarian tissue. Lane1. Testis cDNA; Lane 2. Negative control; Lane 3. Ovary cDNA. (B) Western blotting results for FAM9B expression in both testes and ovaries. Lane 4. Testis protein; Lane 5. Negative control; Lane 6. Ovary protein. All experiments were repeated and analyzed at least three times. Actin mRNA and protein were used as internal control. All bar graphs show the mean±s.e.m. *P<0.05, **P<0.001.

have been associated with clinical features, including mental retardation, short stature, Kallmann syndrome, and infertility.

### Fam9b transcripts and protein expression in human testes and fetal ovaries

Fam9b transcript was measured using real-time quantitative PCR (qPCR) and RT-PCR in the aborted human testes and ovaries (Fig 2A and 2B). Total RNA was extracted from eight human tissue samples and qPCR analysis was performed. The results showed that Fam9b mRNAs are highly expressed in testes and ovarian tissue.

Fam9b protein was detected in human testes and fetal ovaries using western blotting assays. In addition, western blotting also showed that one specific form of protein with a molecular size of ~20 kDa was present in testes and ovaries (Fig 2C).

### FAM9B may play a role in human spermatogenesis and fetal oogenesis

FAM9B expression analysis was performed on adult testicular sections (Fig 3) using immunostaining. Confocal fluorescence microscopy demonstrated that FAM9B was localized in dense regions that appeared in human germ cell nuclei after immunostaining. Higher expression of FAM9B was found in the spermatocyte nuclei (Fig 3).

FAM9B was detected in human ovaries using immunocytochemical analysis (Fig 4). It was also detected in human fetal ovarian tissue (Fig 4A). FAM9B was distributed in adult ovaries

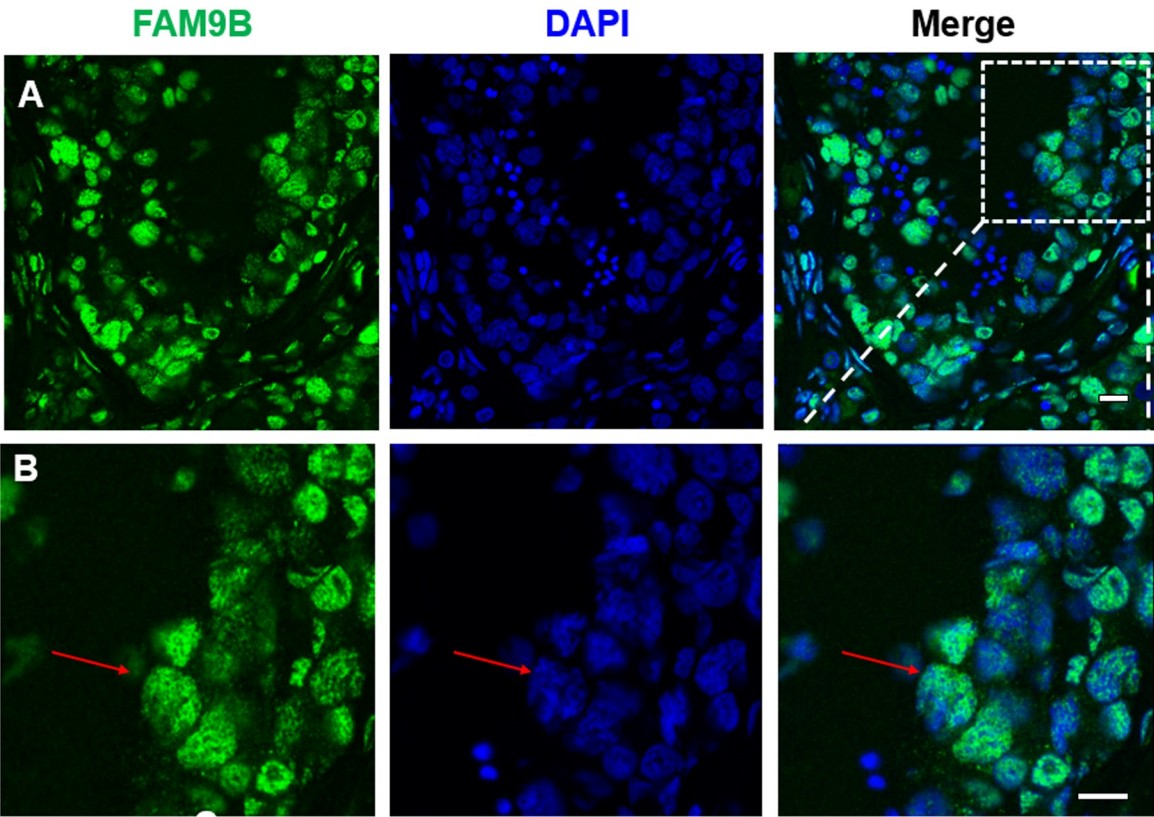

**Fig 3. Immunostaining results for FAM9B expression in human testicular sections.** (A) FAM9B (green) is clearly present in both testicular cell nucleus and cytoplasm, localized in primary spermatocyte nucleus (red arrow), and evident in sertoli cell cytoplasm. FAM9B (green) is localized in the nucleus of germ cells shown by red arrows. Nuclei are stained with DAPI (blue). Bars = 10 µm.

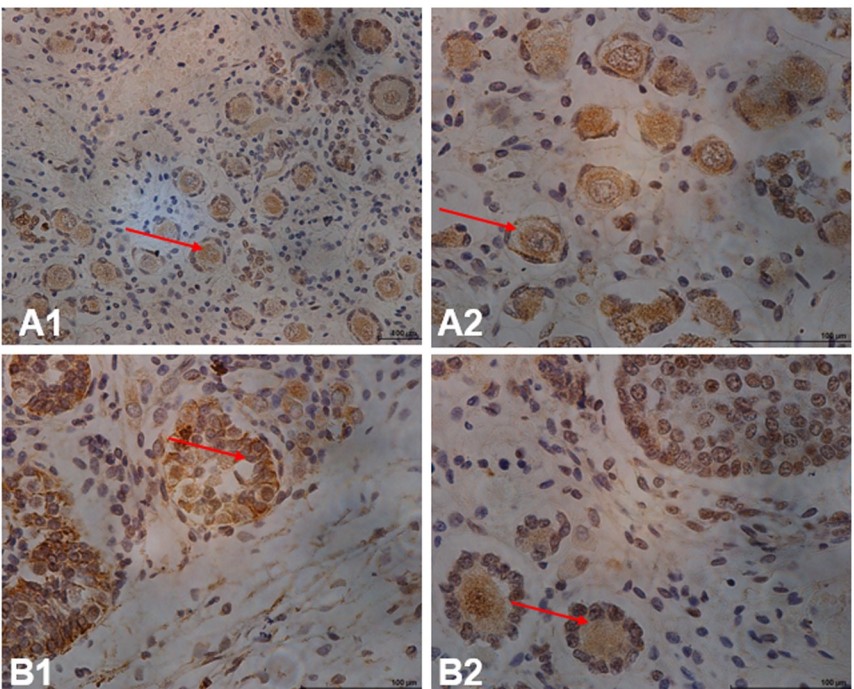

**Fig 4. Immunostaining results for FAM9B expression in human ovarian sections.** In ovary cells, FAM9B was localized in the nucleus and cytoplasm. FAM9B is clearly present in ovarian sections (4A). FAM9B is distributed in adult ovaries (red arrow) and localized in granulosa cells (Fig 4B). FAM9B is also evident in follicle cell nucleus and diffusely dispersed in granular cell cytoplasm (red arrow). Bars = 10 μm.

and localized in granulosa cells (Fig 4B). FAM9B was evident in follicle cell nuclei and diffusely dispersed in granular cell cytoplasm. Therefore, FAM9B may play a role in oocyte maturation.

## FAM9B is a novel meiosis-associated protein

FAM9B expression was detected in chromosome spreads of different human spermatocyte cells using immunostaining analysis (Fig 5). FAM9B was localized in the SC region, similar to SYCP3. SYCP3 is a well-known SC marker in the spermatocyte chromosome spreads. Further study investigated whether FAM9B was partly co-localized in SYCP3 and expressed as distinct points along the chromosome axis during meiosis. In leptotene spermatocytes, FAM9B localization matched the SYCP3 staining pattern. In pachytene spermatocytes, SYCP3 staining was restricted to the SC. FAM9B and SYCP3 may participate in the meiosis stage of spermatogenesis and may be involved in SC formation and meiosis in human germ cells. FAM9B may also perform a function similar to SYCP3 in human SC formation, gametogenesis, and fertility.

## FAM9B is partly co-localized with γH2AX

To confirm that FAM9B was indeed related to meiosis, FAM9B and γH2AX co-localization in the nuclei of primary spermatocytes in meiosis I was evaluated using confocal laser scanning microscope (Fig 6). FAM9B is a nuclear protein that is localized in the spermatocytes in meiosis I and partly co-localized with γH2AX in the spermatocyte nuclei in the first meiotic prophase. It is widely accepted that γH2AX is a novel biomarker that is used to monitor DNA repair in primary spermatocytes during meiosis. The results showed that FAM9B had a

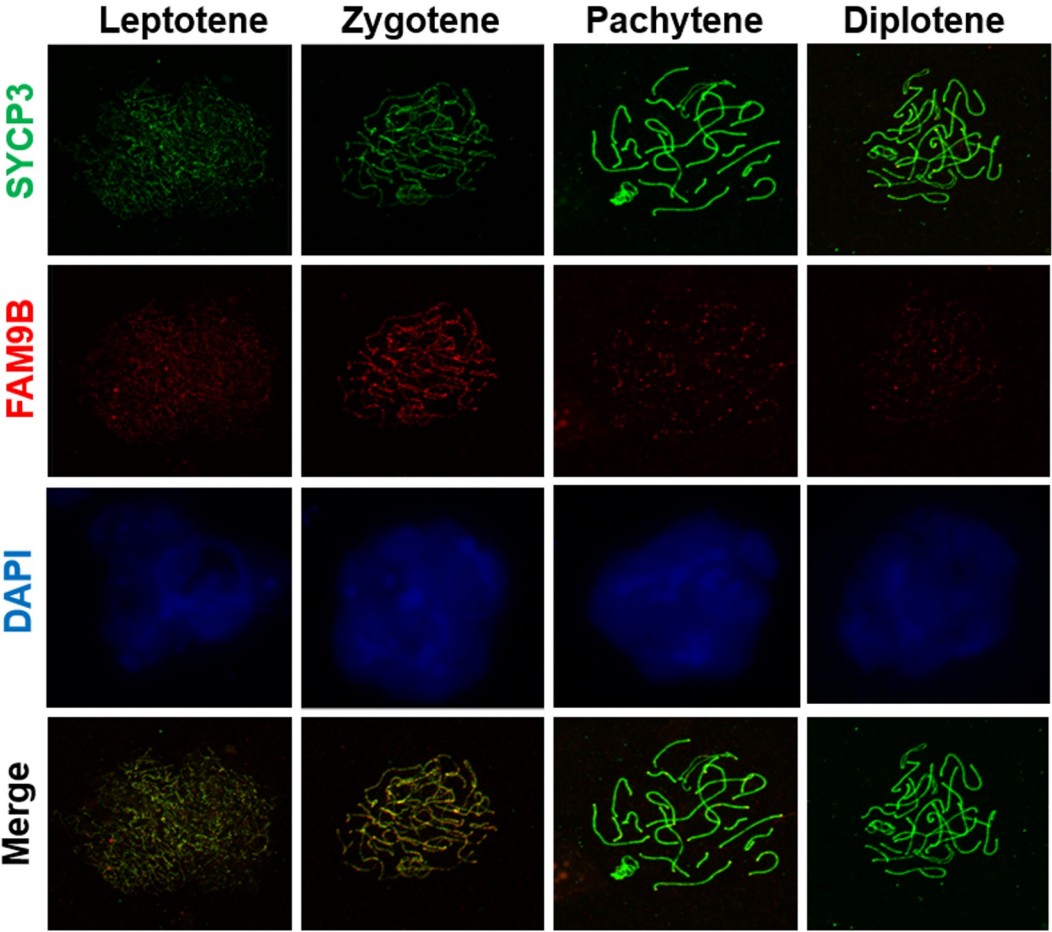

**Fig 5. FAM9B and SYCP3 proteins are partly co-localized in chromosome sections.** FAM9B is localized in the chromosome axis as distinct foci during (A) leptotene, (B) zygotene, (C) pachytene, and (D) early diplotene. SYCP3 were used to marker protein for all stages of meiotic prophase I. FAM9B (green) and SYCP3 (red) are co-localized in SC of chromosome sections in primary spermatocytes. The pictures are representative of three independent experiments. Scale bar = 5 μm.

different distribution pattern than γH2AX. The level of γH2AX was increased in the spermatocyte nuclei during meiosis I. In contrast, the protein level of FAM9B decreased gradually. During the first meiotic division, partial co-localization of FAM9B, SYCP3, and γH2AX in the nuclei of primary spermatocytes may be associated with DNA recombination in meiosis.

## Discussion

Although Fam9b mRNA and protein expression results in human testes have been reported in previous studies [29], some questions remain unanswered, including the location and quantity of FAM9B expression in 8 human tissues. In the present study, FAM9B mRNA and protein levels were detected in the testes and ovaries of fetal tissue using qPCR, RT-PCR, and western blotting assays (Fig 2). Western blotting results revealed that one protein forms in the testes and ovaries, which is recognized by rabbit polyclonal FAM9B antibody. RT-PCR revealed that FAM9B is expressed in the ovarian section follicles. FAM9B plays an important role in male and female meiosis.

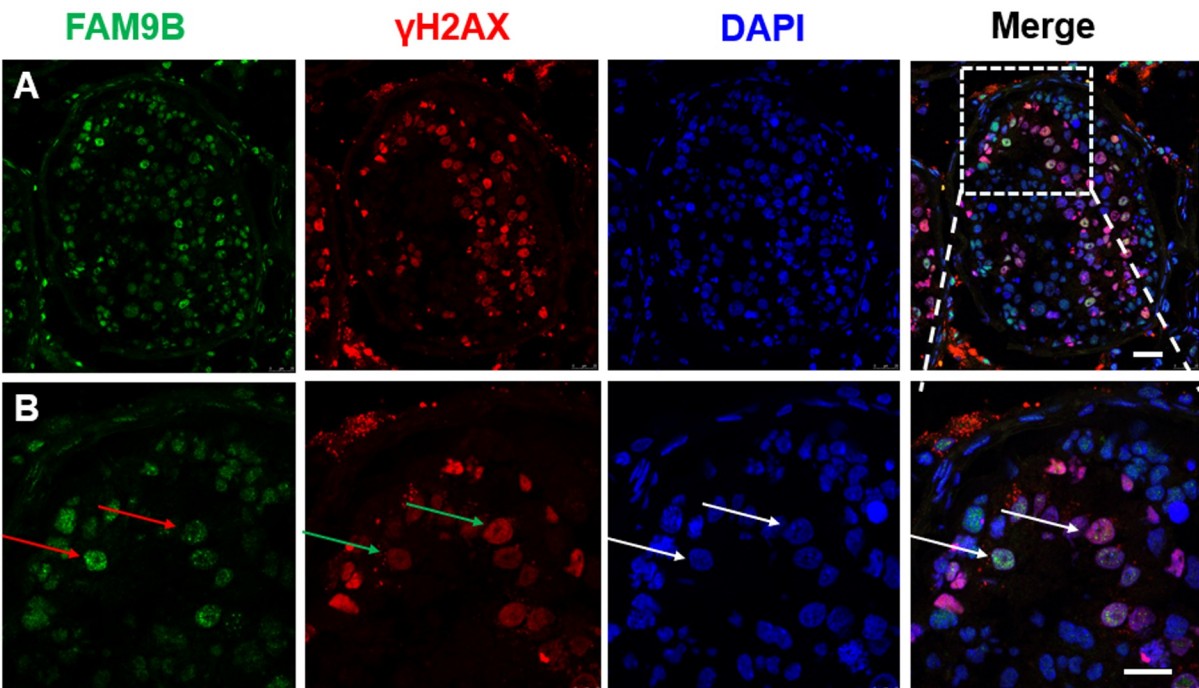

**Fig 6. FAM9B and γH2AX proteins are partly co-localized in spermatocytes.** The presence of γH2AX indicates unsynapsed and unrepaired DSB. FAM9B (green) and γH2AX (red) are co-localized in the nuclear of spermatocytes (red arrow). FAM9B is partly co-localized with γH2AX in the spermatocyte nucleus in the first meiotic prophase (white arrow). They have a similar distribution pattern. Scale bar = 10 μm.

SYCP3 is a major component of the chromosome axes. FAM9B shares similarities with Sycp3, suggesting a similar role in meiosis. To further investigate the subcellular localization of FAM9B, chromosome spreading was performed and FAM9B was exclusively located in spermatocyte nuclei using immunohistochemistry and immunocytochemistry. Although FAM9B is generally localized in the nuclei of primary spermatocytes, it is specifically localized during leptotene and diplotene in spermatocyte nuclei. SYCP3 is a well-known SC marker in the spermatocyte chromosome spreads. FAM9B was localized in the SC region, similar to SYCP3. These results showed that FAM9B is a novel meiosis protein that may participate in SC formation similar to SYCP3.

It is widely accepted that γH2AX is phosphorylated at the DSB sites. γH2AX has been detected during multiple developmental steps in adult germ cells. During meiosis, γH2AX participates in recombination and sex chromosome inactivation in the meiotic processes. In addition, γH2AX is an essential signal for the silencing of unsynapsed sex chromosomes during male meiosis. FAM9B is a nuclear protein localized in the spermatocytes in meiosis I and is partly co-localized with γH2AX in the spermatocyte nuclei in the first meiotic prophase. FAM9B acts as a novel meiosis-associated protein and may have a role in regulation of meiotic processes. It is possible that γH2AX may recruit FAM9B in DSBs and homologous recombination during meiosis in human testes. Deletion in FAM9B may be associated with a number of clinical conditions in males, such as infertility.

In conclusion, FAM9B is abundantly expressed and localized in human testes and ovaries, suggesting that FAM9B is a novel meiosis-associated protein that may play an important role in human meiosis.

## Supporting information

**S1 Raw images.**
(PDF)

**S2 Raw images.**
(PDF)

**S3 Raw images.**
(PDF)

**S4 Raw images.**
(PDF)

**S5 Raw images.**
(PDF)

## Author Contributions

**Conceptualization:** Xin-jie Zhuang, Ping Liu.

**Data curation:** Jun-sheng Li.

**Formal analysis:** Xue Feng, Wen-hao Tang, Xiao-ying Zheng.

**Funding acquisition:** Wen-hao Tang, Rong Li, Jie Qiao.

**Methodology:** Jin-liang Zhu, Xiao-ying Zheng.

**Project administration:** Jin-liang Zhu.

**Resources:** Xue Feng.

**Software:** Ming Li.

**Supervision:** Ming Li.

**Validation:** Xin-jie Zhuang.

**Visualization:** Jun-sheng Li.

**Writing – original draft:** Xin-jie Zhuang.

**Writing – review & editing:** Rong Li, Ping Liu, Jie Qiao.

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
