## [Decision Letter · Decision Letter 0]

16 Jun 2021

PONE-D-21-13955

FAM9B serves as a novel meiosis-related protein localized in meiotic chromosome cores and associated with human gametogenesis

PLOS ONE

Dear Dr. Liu,

Thank you for submitting your manuscript to PLOS ONE. After careful consideration, we feel that it has merit but does not fully meet PLOS ONE’s publication criteria as it currently stands. Therefore, we invite you to submit a revised version of the manuscript that addresses the points raised by the two reviewers during the review process in the following.

We look forward to receiving your revised manuscript.

Kind regards,

Xuejiang Guo, Ph.D.

Academic Editor

PLOS ONE

Journal Requirements:

Reviewers' comments:

Reviewer's Responses to Questions

**Comments to the Author**

1. Is the manuscript technically sound, and do the data support the conclusions?

Reviewer #1: Yes

Reviewer #2: Yes

2. Has the statistical analysis been performed appropriately and rigorously? 

Reviewer #1: Yes

Reviewer #2: Yes

3. Have the authors made all data underlying the findings in their manuscript fully available?

Reviewer #1: Yes

Reviewer #2: Yes

4. Is the manuscript presented in an intelligible fashion and written in standard English?

Reviewer #1: Yes

Reviewer #2: Yes

5. Review Comments to the Author

Reviewer #1: 1、 Is the manuscript technically sound, and do the data support the conclusion？

This manuscript focused on human gametogenesis which is a highly orchestrated process and involves numerous genes. The authors studied Fam9b, which shows high homology with the SC protein SYCP3. The RT-PCR and western blot results showed Fam9b was expressed in

Human testis and ovary at mRNA and protein level. While the IM results revealed the distribution pattern of Fam9b during spermatogenesis and oogenesis. Especially, FAM9B partly co-localized with SYCP3 and γH2AX which indicates the potential function of FAM9B in SC formation and DNA recombination. The design of the article is reasonable and the results can support the conclusion. But some details should be added, such as the replication of the experiments, the negative control of Fig2B. And the information of γH2AX should be added in Introduction part since it is very important for the conclusion of FAM9B in meiotic recombination.

2、 Has the statistical analysis been performed appropriately and rigorously?

The qPCR was carried out to study the mRNA level of FAM9B in different tissues which showed that FAM9B was highly expressed in reproductive organs. The figure legends should include the information on sample type, replication and so on.

3、 Have the authors made all data underlying the findings in their manuscript fully available?

The data showed in this manuscript is complete and could support the results. But in the figure legends, the descriptive information should be added such as the sample type, the replication, the color and arrows information.

4、 Is the manuscript presented in an intelligible fashion and written in standard English?

The article is well written and easy to understand, but there are still some sentences that need to be revised. Such as:

LINE19: has been found

Line 20: delete” the ”

Line21: reported previously

Line25: co-localized

Line28: indicated

Line 98: samples

Line 111: blocked in TBST ?

LINE 145: Fam9b may have functions as SYCP3 in spermatogenesis.

Line161: expression---distribution or localization ?

Line 170: meiosis

Line 183: co-localization?? Or only to explore the localization pattern of Fam9b and γH2AX.

Reviewer #2: In this manuscript, Zhuang et al. explored the expression pattern and localization of a novel protein FAM9B that may play a role in meiosis. It is interesting to note that this protein has a similar distribution pattern to the localization of Sycp3 and γH2AX. Overall, this work showed good data quality and revealed FAM9B may offer a new perspective for understanding the molecular mechanisms involved in meiosis of human gametogenesis.

It is better to further analyze the co-localization of FAM9B and γH2AX on spermatocyte spreading slides, to provide more accurate evidence for whether FAM9B is a bio-marker of DSB.

Spelling and grammatical checking is needed before publication.

6. PLOS authors have the option to publish the peer review history of their article (what does this mean?). If published, this will include your full peer review and any attached files.

Reviewer #1: **Yes: **Mengcheng Luo

Reviewer #2: No

---

## [Author Response · Author response to Decision Letter 0]

22 Jul 2021

Dear reviewers,

 We appreciate it very much for your helpful suggestions. We re-check and modified the MS including symbols or characters. And the modified manuscript was re-edited by International Science Editing. 

 We are pleased to let you know that we have finished revision of the manuscript according to the suggestions by you. The point-by-point responses are as the following. Please let me know if any further improvement is needed. Thank you very much for your kind help!

Sincerely yours,

Dr. Ping Liu

Center for Reproductive Medicine, Department of Obstetrics and Gynecology, 

Key Laboratory of Assisted Reproduction,Ministry of Education,

Peking University Third Hospital; Beijing 100191, PR China

E-mail: Bysylp@sina.com

---

## [Decision Letter · Decision Letter 1]

27 Aug 2021

FAM9B serves as a novel meiosis-related protein localized in meiotic chromosome cores and is associated with human gametogenesis

PONE-D-21-13955R1

Dear Dr. Liu,

We’re pleased to inform you that your manuscript has been judged scientifically suitable for publication and will be formally accepted for publication once it meets all outstanding technical requirements.

Kind regards,

Xuejiang Guo, Ph.D.

Academic Editor

PLOS ONE

Additional Editor Comments (optional):

Reviewers' comments:

Reviewer's Responses to Questions

**Comments to the Author**

1. If the authors have adequately addressed your comments raised in a previous round of review and you feel that this manuscript is now acceptable for publication, you may indicate that here to bypass the “Comments to the Author” section, enter your conflict of interest statement in the “Confidential to Editor” section, and submit your "Accept" recommendation.

Reviewer #1: All comments have been addressed

Reviewer #2: All comments have been addressed

2. Is the manuscript technically sound, and do the data support the conclusions?

Reviewer #1: Yes

Reviewer #2: Yes

3. Has the statistical analysis been performed appropriately and rigorously? 

Reviewer #1: Yes

Reviewer #2: Yes

4. Have the authors made all data underlying the findings in their manuscript fully available?

Reviewer #1: Yes

Reviewer #2: Yes

5. Is the manuscript presented in an intelligible fashion and written in standard English?

Reviewer #1: Yes

Reviewer #2: Yes

6. Review Comments to the Author

Reviewer #1: (No Response)

Reviewer #2: The authors have well addressed all my comments. There is no further comment. Thanks for an interesting paper.

7. PLOS authors have the option to publish the peer review history of their article (what does this mean?). If published, this will include your full peer review and any attached files.

Reviewer #1: No

Reviewer #2: No

---

## [Editor Report · Acceptance letter]

1 Sep 2021

PONE-D-21-13955R1 

FAM9B serves as a novel meiosis-related protein localized in meiotic chromosome cores and is associated with human gametogenesis 

Dear Dr. Liu:

I'm pleased to inform you that your manuscript has been deemed suitable for publication in PLOS ONE. Congratulations! Your manuscript is now with our production department. 

Kind regards, 

on behalf of

Prof. Xuejiang Guo 

Academic Editor

PLOS ONE